# What Happens in Male Dogs after Treatment with a 4.7 mg Deslorelin Implant? II. Recovery of Testicular Function after Implant Removal

**DOI:** 10.3390/ani12192545

**Published:** 2022-09-23

**Authors:** Sabrina Stempel, Hanna Körber, Larena Reifarth, Gerhard Schuler, Sandra Goericke-Pesch

**Affiliations:** 1Reproductive Unit, Clinic for Small Animals, University of Veterinary Medicine Hanover, Foundation, 30559 Hanover, Germany; 2Clinic for Obstetrics, Gynecology and Andrology of Large and Small Animals, Klinikum Veterinärmedizin, Justus-Liebig-University Giessen, 35392 Giessen, Germany

**Keywords:** deslorelin, GnRH stimulation test, hCG stimulation test, male dogs, restart, recrudescence, hemicastration

## Abstract

**Simple Summary:**

Although widely used in clinical practice as an alternative to surgical castration, the recovery of testicular endocrine and germinative function subsequent to treatment with a 4.7 mg deslorelin implant has not been well characterized until now. It is, however, relevant for the veterinary practitioner to advise the pet owner about the expected return to “normal” testosterone concentrations, prostatic volume, and pre-treatment semen quality. Our study showed that testosterone was no longer different from untreated controls on D14 after implant removal, whereas normospermia was reached on D84-133. All of the effects induced were fully reversible, albeit after individually varying time periods.

**Abstract:**

Although deslorelin slow-release implants are widely used in the clinic, detailed published information about the recovery of testosterone concentrations (T), semen quality, and testicular and prostatic volume (TV, PV) after treatment is still missing. This article aims to characterize changes during restart after a five-months treatment and subsequent implant removal. Seven male Beagle dogs were treated with deslorelin (treatment group, TG), and three saline-treated dogs served as controls (CG). Deslorelin implants were removed after five months (D ex), followed by detailed andrological examinations for TV, PV, semen collection, and blood sampling for T-analysis with/without GnRH/hCG stimulation tests. TV, PV, and T increased rapidly after D ex in TG, not differing from CG from D91 (TV), D49 (PV), and D14 (T). The first sperm-containing ejaculates were collected between D49 and 70, whereas the samples were normospermic between D84 and 133. A T increase (>0.1 ng/mL) subsequent to the GnRH/hCG stimulation test was observed from D28/29 onwards, respectively. Histological assessment of testicular tissue at the end of the observational period (D149 after implant removal) revealed normal spermatogenesis. Our data confirm that the restart of endocrine and germinative testicular function is highly variable, but nevertheless, all of the effects induced were reversible.

## 1. Introduction

Reversible alternatives to surgical castration are frequently demanded in pets [1,2,3,4,5,6,7,8], and the use of deslorelin slow-release implants (SRI) is a registered and suitable method in male dogs [3,5,6,9].

Whereas the initial period subsequent to treatment with a 4.7 mg deslorelin implant was recently characterized in detail [10], until now, the restart and recovery of endocrine and germinative testicular function following treatment are not well characterized. Whereas very few studies are available in dogs, all either referring to the 6 mg deslorelin [11] or to other GnRH agonist SRIs [2,12,13], indicating the full reversibility of treatment-induced effects on testicular endocrine and germinative function, some information about the reversibility of 4.7 mg deslorelin induced effects is available in male and female cats [14,15,16,17,18,19]. Treatment-induced effects in male cats were confirmed to be reversible after the end of efficacy [16] but also after four [19] or three, six, and nine months of treatment and subsequent implant removal [17], respectively. Interestingly, the time to restoration of serum testosterone was, on average, three to four weeks and consequently similar in cats after the spontaneous end of effect (2.9 ± 1.1 weeks to >0.5 ng/mL; 3.7 ± 1.7 weeks to >1 ng/mL) [16] as well as after implant removal (23 ± 6 days three/six months, 22 ± 7 days nine months of treatment; testosterone >0.1 ng/mL) [17]. Ferre-Dolcet et al. [17], however, also reported about one “outlier” cat where it took 231 days after removal subsequent to three months of deslorelin treatment to return to “normal” serum testosterone concentrations. This long duration was ascribed to the breaking of the implant. Similar to testosterone , testicular volume was restored, namely 6.9 ± 3.4 weeks after the last basal testosterone [16]. Variable spermatogenesis, but no spermatozoa were found at orchiectomy two weeks after the first identification of testosterone >0.1 ng/mL [17]. Different to this compared to pre-treatment, Novotny et al. [19] identified higher total sperm counts already one month after implant removal and histologically elongated spermatids one, two, and three months after a 4-month treatment with a 4.7 mg deslorelin SRI. Finally, complete spermatogenesis was observed 13.6 to 47.6 weeks after the last basal testosterone in all seven toms at the time of castration and return to fertility in 4/4 tomcats that were allowed to mate queens supporting full reversibility in terms of treatment-induced effects [16].

Despite being licensed for over a decade in male dogs, all of these data are missing for male dogs. This clearly indicates the need for an investigation of the return to pre-treatment testosterone concentrations, the effect on the testicular and prostatic volume, and the recovery of semen quality in the male dog subsequent to treatment with a 4.7 mg deslorelin SRI. To fill this gap and to provide this highly relevant information to pet owners and veterinarians, we conducted this study to gain detailed insights into the testicular endocrine and germinative restart following deslorelin treatment and subsequent implant removal. The parameters assessed include testosterone concentrations, testicular and prostatic volume, as well as ejaculate parameters, and testicular histomorphology.

## 2. Materials and Methods

Animal experiment approval was authorized by the respective authorities (AZ 19/3203, LAVES Hannover). The study approved and performed aimed to investigate the effects induced by a 5-month treatment with a 4.7 mg deslorelin implant and subsequent implant removal. The reason for choosing five instead of six months (the manufacturer’s claimed duration of effect) as the time point of implant removal was related to our previous studies [20,21] and unpublished data using a deslorelin SRI showing that dogs were fully downregulated after five, but not necessarily after six months. This was necessary as we aimed for a homogenous starting point to provide more generally valid data, similar to what was provided for cats removing the SRI at various time points [17]. However, unfortunately during the study, it became obvious that despite this choice of a five-month treatment period, one dog had already started reverting (see later). At this time, changes to the experimental design were impossible due to animal experimentation approval.

We recently presented the results of the first five months after treatment until implant removal [10], whereas the results obtained after implant removal are included in the current paper. All of the methods were described in detail recently [10].

### 2.1. Animals and Experimental Design

This study included a total number of ten mature male beagle dogs. The study was designed as a randomized controlled, blinded laboratory trial. Seven healthy male beagle dogs were treated with a 4.7 mg deslorelin SRI (Suprelorin^®^ 4.7 mg; Virbac, Carros, France) in the paraumbilical area over five months representing the group (TG, *n* = 6 normospermia; *n* = 1 pathospermia with moderate oligozoospermia and low-grade teratozoospermia). Three dogs were injected with sterile saline, representing the control group (CG, *n* = 3; normospermia). Implant removal was performed under general anesthesia, and the dogs were hemicastrated (right testis) on the same day (D ex) for the evaluation of testicular morphology at the end of the treatment period. The andrological examinations in TG and CG dogs performed during the observational period covering this part of the study, including the testicular measurements and the ultrasound examinations of the prostate gland, semen collections, and blood samplings for testosterone, are presented in detail in Figure 1.

On D149 after implant removal, at the end of the observational period, the left testis was surgically removed under general anesthesia and further processed for histological evaluation. One dog from the TG (Lui) had testosterone concentrations within the physiological range and spermatozoa in the semen sample before implant removal, indicating that the effects on testicular endocrine and germinative function were already reversed. Therefore, this dog was excluded from further analysis, and the data of *n* = 6 dogs from the TG were included.

### 2.2. Physical and Andrological Examination, Ultrasound of the Prostate Gland

Over the course of the study, the clinical examinations, including body weight and clinical parameters, were performed regularly, namely daily from D ex up to D7 and every two weeks after that until the end of the observation period. Andrological examinations, including the measurements of testicular dimensions and sonographic measurements of the prostate size, were performed on a weekly basis (D7, 14, 21, …, 140, 147), as described. The testicular volume (cm^3^, TV) and the prostate volume (cm^3^, PV) were calculated by the formula of an ellipsoid (length × width × height × 0.523) [22,23,24,25]. All of the examinations were performed by the same blinded person (first author) to avoid operator-related variance [26].

### 2.3. Semen Collection and Analysis

As the aim was to confirm the reversibility of the suppressed testicular germinative function, namely the return to normospermia according to the reference range described earlier [27,28], semen collection was attempted weekly after D ex starting on D14 until D147 either in the presence of a bitch in heat or an anestrus bitch with an additional cloth with vaginal discharge from an estrus bitch. All of the dogs were trained and had been regularly collected before being included in the study. The investigated parameters were the total ejaculate volume (VOL), the percentage of progressively motile (% PM), of viable (% VS), and morphologically abnormal spermatozoa (% MAS), as well as the total sperm count (×10^6^).

### 2.4. Blood Collection, GnRH/hCG Stimulation Tests and Hormone Analysis

For the testosterone analysis, blood (2 mL plasma) was collected by venipuncture weekly from D ex until D56 and also on D149 at the same time of the day. The samples were centrifuged (Heraeus Megafuge 2.0 R, Landgraf Laborsysteme HLL GmbH, Langenhagen, Germany), the plasma was aliquoted, frozen, and stored at −80 °C until analysis. Testosterone was analyzed using an established radioimmunoassay (RIA) [29,30] with a lower detection limit of 0.05 ng/mL. The intra- and interassay-coefficient were between 3.7% and 7.6%, respectively. Testosterone concentrations (T) of ≤0.1 ng/mL were considered as basal [13,21,31].

Additionally, GnRH stimulation tests (0.4 µg/kg buserelin acetate, Receptal^®^ IV, Intervet Deutschland GmbH, Unterschleißheim, Germany) [8] were performed, followed by hCG stimulation tests (750 IU/dog hCG, Ovogest^®^ IM, Intervet Deutschland GmbH, Unterschleißheim, Germany) [8,32] on a subsequent day, namely on D7/8, 28/29, and 56/57 after implant removal. All of the pre- and post-stimulation samples (“before”/“after”) were collected at the corresponding times of the day, with the post-stimulation samples being collected one hour after buserelin acetate or hCG injection. GnRH/hCG stimulation tests aimed to verify the functionality of the hypothalamus–pituitary–gonadal axis (buserelin) and Leydig cells (hCG).

### 2.5. Removal of the Testes, Processing of Testicular Tissues and Histology of Testes

On D149, the left (second) testis was surgically removed under general anesthesia in all dogs by the senior author. The dogs were premedicated with dexmedetomidine 5 µg/kg (CP-Pharma, Burgdorf, Germany) and levomethadone 0.2 mg/kg (MSD, München, Germany), followed by the administration of propofol 2–4 mg/kg (CP-Pharma, Burgdorf, Germany), and ketamine 1 mg/kg (CP-Pharma, Burgdorf, Germany) maintained by isoflurane (1%). Immediately after surgery (D149), the testes were trimmed in pieces of approximately 1 cm^3^ and fixed in Bouin’s solution. Following washing with 70% ethanol and embedding in paraffin, the sections were cut, hematoxylin–eosin stained, and mounted with Roti^®^ Histokitt (Carl Roth, Karlsruhe, Germany). A histological assessment of spermatogenesis was performed at 400× magnification (Zeiss West, Oberkochen, Germany [20]).

### 2.6. Statistical Analyses

Statistical analysis was performed using GraphPad Prism (GraphPad Prism Software 8.3.0, San Diego, CA, USA). Data collection and presentation were performed with Microsoft Excel (2019, Microsoft Corporation, Redmond, WA, USA). The data of all dogs (TG: *n* = 6, CG: *n* = 3) were used for analysis, except for the statistical analysis of the semen parameters. Due to low volume, not all of the ejaculate parameters could be analyzed on all of the collection days in the TG. Thus, only VOL was analyzed statistically over the complete observational period. In addition to the raw data, the results from the GnRH/hCG stimulation test were presented as a relative increase after stimulation, with “T pre-stimulation” considered as 100% and “T-post-stimulation” as the % change.

The Shapiro–Wilk test was used to determine whether the data were normally distributed. The normally distributed data (TV) was given as mean with standard deviation (mean ± SD). The unevenly distributed data (PV, semen parameters) were presented as median and first/third quartile (Q1/Q3) or were given as geometric mean and dispersion factor [$\bar{x_{g}}$ (DF)] (T). For comparative reasons and for better illustration, the PV and semen data were also presented as median (Q1/Q3) if the distribution of one parameter at a specific time point was normally distributed when the others were unevenly distributed. In addition, for further statistical analysis, T values below the detection limit of the RIA (<0.05 ng/mL) were defined as 0.025 ng/mL.

The results of the examinations before implant application (“prior to treatment”) were considered to be “D0”. The results of the last examination day before the implant removal were considered to be “downregulation”. As described in manuscript I, these results were obtained 147 days after treatment for PV, semen analysis, and GnRH/hCG stimulation tests and 154 days after treatment for TV, T, and first hemicastration. For simplification in the subsequent paragraphs, the results are presented as “D ex” (equivalent to “before implant removal”). Similarly, the results representing “recovery” were obtained on D147 after implant removal for TV, PV, and semen analysis and on D149 for T and second hemicastration.

The aim of this study was to Identify the differences within the groups over time (TG/CG) and between the groups TG and CG. Additionally, the effect of the treatment was specifically studied in TG, comparing the results at the following three time points: 1. Prior to treatment (D0), 2. Under the full effect of treatment (“downregulation”, see “D ex” above, five months after treatment before removal), and 3. After recovery. For the ejaculate parameters in the normospermic dogs from the TG (*n* = 5), a comparison was made between 1. Prior to treatment (D0, pre-treatment), 2. First assessable ejaculate, and 3. First normospermic ejaculate.

To identify significant differences over time within one group (TG/CG), a repeated measures ANOVA was performed in the case of normal data distribution (CG: VOL, PM, VS, MAS, TV, hCG; TG: TV) and the Friedman-test (paired, non-parametric) in case of no normal distribution (CG: TSC, PV, T, GnRH; TG: VOL, PV, T, GnRH, hCG). The same tests were applied when comparing D0, D ex, and “recovery”, followed by Dunn’s multiple comparisons test in the case of significant differences being (*p* < 0.05) indicated.

Aiming to identify when the results between TG and CG no longer differed, the results from corresponding examination days were compared using parametric, unpaired *t*-tests or non-parametric, unpaired tests (Mann–Whitney test) depending on the data distribution.

Aiming to identify when the results within one group (TG or CG) did no longer differ from the prior-to-treatment results (D0), a paired t-test or Wilcoxon signed-rank test was performed for the respective parameter depending on the data distribution.

All of the results with *p* ≤ 0.05 were considered statistically significant.

## 3. Results

### 3.1. Physical and Andrological Examination

All of the male dogs were clinically healthy during the whole observation period. 

The mean absolute TV increased from 10.1 ± 1.2 cm^3^ on D ex to 16.0 cm^3^ ± 2.1 cm^3^ on D147 in CG and from 2.9 cm^3^ ± 0.5 cm^3^ to 13.9 cm^3^ ± 2.0 cm^3^ in TG, indicating a relative increase by a factor of 1.6 in CG and 4.8 in TG. Statistical analysis confirmed a significant increase in the absolute TV of the left testis over time in both groups (CG: *p* = 0.0239, TG: *p* < 0.0001) (Figure 2). Whereas TV of TG was decreased compared to CG on D ex (*p* < 0.0001), a pairwise comparison revealed that TV in TG was no longer different from CG from D91 onwards.

Comparing the results from those prior to treatment (D0) to those at the end of the observation period, TV increased by 1.6 times in TG and 1.5 times in CG, likely as a consequence of hemicastration.

### 3.2. Ultrasound of the Prostate Gland

The absolute PV varied in median between 9.7 cm^3^ (Q1: 8.9 cm^3^; Q3: 9.8 cm^3^) on D ex and 12.2 cm^3^ (Q1: 11.5 cm^3^; Q3: 12.2 cm^3^) at the end of the study (D147) in CG and between 1.8 cm^3^ (Q1: 1.7 cm^3^; Q3: 1.9 cm^3^) and 11.3 cm^3^ (Q1: 8.2 cm^3^; Q3: 12.0 cm^3^) in TG with the change over time only being significant in TG (*p* < 0.0001), but not CG (*p* = 0.079) (Figure 3). PV was significantly smaller in TG compared to CG on D ex (*p* < 0.0001) but did not differ between TG and CG from D49 onwards. Although PV increased by a factor of 6.2 from D ex to the end of the observational period in TG, the final PV was neither different from the prior-to-treatment PV in TG (D0) nor from PV in CG at the end of the observation (*p* = 0.3809). Whereas the sonographic appearance of the prostates of the TG animals was of inhomogeneous echogenicity and had a more circular shape on D ex, the typical “butterfly” shape was restored over time, as well as the homogeneous appearance.

### 3.3. Semen Collection and Analysis

The libido returned to normal between D21 and D63 in all of the TG dogs. As stated above, only the dogs that had normospermic ejaculates at the start of the study and no spermatozoa in the last sample before implant removal were included in the statistical data analysis. One dog (Murphy), showing teratozoospermia, is described separately (see below). Complete semen analysis was impossible on some occasions in TG due to the low ejaculate volume and sperm counts. For this reason, all of the parameters of TG, except for VOL, are presented descriptively (Figure 4, Figure 5, Figure 6, Figure 7 and Figure 8).

In TG, the first spermatozoa were obtained between D49 and 70 after implant removal, whereas the first complete ejaculate analysis was possible between D63 and 91. Expectedly, the semen quality changed variably over time, with VOL and TSC increasing over time. Whereas the percentages of PM and VS barely changed, MAS was increased in the first samples after implant removal but decreased from D112 onwards to approximately 10% at the end of the observation period. Normospermia (5/5) was reached between D84 and 133. Details about the recovery of semen quality by examination day are provided in Appendix A.

In CG, the libido was not affected by hemicastration, with libido being high in all three dogs at the first semen collection after surgery (D14). Only TSC, but not ejaculate volume, PM, VS, and MAS, was significantly affected by hemicastration, with the counts being reduced below the reference range for the respective body weight of dogs (<500 × 10^6^); [27,28]. TSC subsequently increased significantly (*p* = 0.0127), and the ejaculates fulfilled the normospermia criteria [27,28] again D21 (*n* = 2) and D70 (*n* = 1) after D ex, respectively.

Considering the dog with altered semen quality prior to treatment (Murphy), the first sperm were identified on D49. Changes over time in this dog are included in Figure 4b, Figure 5b, Figure 6b, Figure 7b and Figure 8b. VOL and TSC increased over time, but TSC only reached 178 × 10^6^ spermatozoa at the end of the study. In total, the semen quality was restored to pre-treatment levels, with oligozoo- and teratozoospermia still being present after treatment and hemicastration.

### 3.4. Testosterone Concentrations and Stimulation Tests

Whereas testosterone in the CG animals varied between 0.3 and 3.8 ng/mL on D ex, two of the six TG dogs had basal testosterone concentrations (≤0.1 ng/mL), three had slightly suprabasal levels (0.1–0.3 ng/mL), and the remaining dog had a testosterone concentration within the physiological range of an intact dog (1.8 ng/mL). Testosterone increased subsequently in all of the TG dogs over time (*p* < 0.0001) and was >0.1 ng/mL on D14 and >0.5 ng/mL on D28, with concentrations not differing from CG from D14 onwards until the end of the study (Figure 9). On the contrary, testosterone was not affected by time in CG (*p* = 0.2371).

Comparing the pre-treatment values to the results at the end of the observation period, no significant differences were identified in both groups (CG: *p* = 0.1844, TG: *p* = 0.3439), indicating the full reversibility of treatment and no impact of hemicastration.

To test for a functional hypothalamus–pituitary–gonadal axis, GnRH stimulation tests (using buserelin) were performed five months after treatment and on D7, 28, and 56 after implant removal and hemicastration.

In TG, the pre-stimulation testosterone concentrations were basal in five of the six treated dogs after five months of deslorelin treatment and in three of the six dogs on D7 after implant removal. Of those animals, two and one dog, respectively, did not respond to buserelin stimulation, whereas an increase was observed in the remaining. On D28 and 56, all of the pre- and post-stimulation testosterone concentrations were above the basal level, and all of the animals (except one TG dog on D28) responded to stimulation. Testosterone increased by a factor of 1.1 to 5.0 in TG (mean 2.1) and by 2.1 to 6.9 in CG (mean 3.7). As clearly visible, the minimum increase was lower in TG than that in CG (at least 2×), namely in five of the six dogs on D28 and four of the six on D56, it was decreased, possibly indicating a prolonged effect on pituitary GnRH receptors. The changes over time were not significant in both groups (CG: *p* = 0.6076; TG: *p* = 0.512) [$\bar{x_{g}}$ (DF); Figure 10].

To test for Leydig cell functionality, hCG stimulation tests were performed in TG and CG five months after deslorelin implant treatment and on D8, 29, and 57. Pre-stimulation testosterone concentrations with hCG were basal (≤0.1 ng/mL) in six out of the six dogs after five months of treatment (TG) and on D8, but in none of the dogs on D29 and D57. Interestingly, five of the six dogs did not respond to hCG stimulation on D ex and D29, and four of the six dogs on D8, even though the pre-stimulation testosterone was >0.1 ng/mL in all respective dogs on D29. Similarly, one of the six dogs did not show an increase in testosterone on D57 (Figure 11).

The response to hCG was also variable in CG: One dog did show a response on all occasions, one showed only a response on D56, and the last dog on D8 and D29. In contrast to CG, the relative testosterone increase in TG changed significantly over time (CG: *p* = 0.4615; TG: *p* = 0.0233). The mean relative increase after stimulation on all study days was 2.3 in CG and 1.1 in TG.

### 3.5. Histology of Testes

Whereas spermatogenesis was arrested on the level of spermatogonia and/or spermatocytes on D ex, the histological assessment revealed fully elongated spermatids in all dogs at the end of the observation period (CG and TG, Figure 12), indicating that spermatogenesis was fully restored, as also shown earlier regarding the results of semen analysis.

### 3.6. Effect of Treatment with a 4.7 mg Deslorelin Implant and Reversibility

Comparing D0, D ex, and recovery, the treatment had a significant effect on TV (*p* < 0.0001), PV (*p* = 0.0001), and T (*p* = 0.0289) (Table 1).

Comparing the ejaculate parameters prior to treatment (D0), the first assessable ejaculates after implant removal and the first normospermic ejaculates from the normospermic dogs (*n* = 5) in TG, MAS (*p* = 0.0239) and TSC (*p* = 0.0361) differed significantly, with the lowest TSC and highest MAS in the first assessable ejaculates (Table 2). In addition, a trend was observed for PM (*p* = 0.0628), with the highest PM observed after recovery.

## 4. Discussion

In one TG dog (Lui), the testosterone concentrations were in the physiological range, and spermatozoa were present in the semen sample five months after the implant application, indicating the end of efficacy at this point in time and the complete reversibility of the effects caused by the implant. For this reason, this dog was excluded from the statistical analysis. These results clearly confirm that in some dogs, the duration of the effect can be shorter than the predicted six months. This is important for the veterinary practitioner when advising clients. Although the current study included only Beagle dogs, a medium-sized breed, and the results cannot be extrapolated for all breeds, or weight classes, our clinical experience with various breeds and body weights of dogs is in good agreement with the observations made. Consequently, the current study provides essential data indicating, however, the need for large-scale (retrospective) studies, including dogs of different breeds, body weights, and ages.

### 4.1. Testicular and Prostatic Volume

Although two studies describe the downregulating effects on semen quality and testosterone after a 4.7 mg deslorelin implant in healthy dogs [3] and dogs with benign prostatic hyperplasia [33], there are no data about the reversibility after a 4.7 mg deslorelin treatment regarding changes in TV. We showed that downregulation caused a significant reduction in TV. As clearly visible, the TV in TG did not differ from CG from D91 (week 13) after implant removal onwards, indicating the reversibility of the deslorelin-induced effects, similar to what was described earlier after an implant with 6 mg deslorelin [11]. The increase in TV observed in both groups at the end of the study compared to the period prior to treatment is likely a consequence of hemicastration. By now, an increase in TV (by a factor of 1.25) was only described following prepubertal hemicastration [34], but not for adult male dogs. Similarly, the effects on PV, namely a significant reduction related to treatment, were also reversible with an increase observed from D28 (week 4) and no further difference in CG from D49 (week 7), similar to previous observations (6 mg deslorelin, [11]). In addition, PV did not differ at the end of the observational study compared to prior to treatment.

### 4.2. Testosterone

Testosterone is crucial for libido and spermatogenesis. Whereas the flare-up and downregulation of testicular endocrine function is well described in the first part of this study, recovery after five months of treatment with a 4.7 mg deslorelin implant and removal has not been described yet. Other than expected, only two of six dogs had basal testosterone concentrations (≤0.1 ng/mL as described earlier, [13,21,31]) on D ex, whereas three dogs had slightly suprabasal values between 0.1 and 0.3 ng/mL, and one even had physiological concentrations (1.8 ng/mL), indicating a shorter duration of downregulation in some dogs. The recovery and restorage of testosterone secretion were rapid and even faster than previously reported after treating dogs with deslorelin and azagly–nafarelin containing slow-release implants [11,31], but similar to what has been recently reported for the tom cat subsequent to removal of the 4.7 mg deslorelin implant [17]. The importance of a shortened duration of effect with single dogs having fully restored testicular function already after five months was emphasized before.

As described before, the analysis of single T samples is not considered optimum because of the pulsatile secretion of T by pulsatile GnRH release [35,36,37]. Our own data with one of the control dogs (Piet), having T of 0.1 ng/mL on D149 (week 21), but normospermic ejaculates, normal libido, and T within the physiological range on all other sampling days, clearly confirm the suitability and need of repeated samplings. To overcome this problem, we performed GnRH/hCG stimulation tests to examine when hypothalamus–pituitary–gonadal axis (GnRH) and Leydig cell (hCG) function are restored. Our study is the first evaluating functionality of pituitary-gonadal-axis and Leydig cells following a five-month treatment with a 4.7 mg deslorelin implant and subsequent implant removal in male dogs. Interestingly, even though T concentrations pre- and post-GnRH stimulation were in the physiological range from D28 onwards in TG, one dog did not respond to stimulation on D28 (Murphy). The trend for lower testosterone increase after buserelin administration in TG (1.1–5.0) compared to CG (2.1–6.9) might be due to individual differences or related to an incompletely restored hypothalamus–pituitary–gonadal axis in some dogs. For the hCG stimulation test, T concentrations were basal pre- (6/6 dogs) and post- (5/6 dogs) administration on D8 but >0.1 ng/mL (or even in the physiological range) pre-stimulation on the other sampling days. Despite this, five of the six dogs did not respond to hCG stimulation on D29 and one dog on D57, either related to still reduced Leydig cell function or due to antibody formation against hCG, as described in mares [38,39], cows [40], bulls [41], queens [42,43], and dogs [44]. The fact that T concentrations tended to be lower on a subsequent day after GnRH stimulation compared to hCG stimulation was also observed earlier in this study in week 21 after treatment with 4.7 mg deslorelin (D147 and 148). As both samplings and stimulation tests were performed at the same time point of the day (morning), further investigations are needed to clarify if there is an underlying reason for the difference.

### 4.3. Semen Evaluation

Whereas the upregulation of endocrine testicular function was quick, the restart of germinative function began later and was more variable, with the first spermatozoa detectable between D49 and 70 (week 7–10) after D ex and first normospermic ejaculates collected between D84 and 133 (week 12–19), according to the reference values [28]. Even though the data of recovery subsequent to treatment with 4.7 mg deslorelin are lacking, first spermatozoa after treatment with a 6 mg deslorelin implant were found approximately five weeks after the last basal testosterone, and the semen quality was normal 11 weeks later [11]. Whereas the occurrence of first sperm was also after 8–12 weeks, following a five-month treatment with 18.5 mg azagly–nafarelin, the return to normospermia was later, namely after 25.6 ± 3.7 weeks [13]. Interestingly, in dogs, the duration of spermatogenesis is described to be 56–63 days [45,46,47], and epididymal sperm maturation takes an average of 10–14 days [48,49], indicating that the restart of spermatogenesis following the abolition of deslorelin treatment is faster for individual spermatozoa, but spermatogenesis in its “full potential” is first restarted after the abolishment of a 4.7 deslorelin SRI after at least one week—coinciding with the recovery of testicular testosterone production.

The observation of high MAS and low TSC in the first ejaculates containing spermatozoa is in good agreement with previous observations with other slow-release implants (6 mg deslorelin: [11], 18.5 mg azagly–nafarelin: [13]). The tendency for improved semen quality in terms of MAS and PM, identified earlier using the azagly–nafarelin implant [13], was only observed for PM in our current study. Interestingly, VS and PM were already high in the first samples indicating that they might be less affected by treatment and recovery than the other semen parameters. In addition, the recovery of testicular function was also confirmed histologically.

Hemicastration in CG did neither affect libido nor any semen parameter, except for TSC, similar to an earlier study [50]. Return to normal TSC within the body weight range [27,28] took three to ten weeks (D21: *n* = 2; D70: *n* = 1), clearly confirming that the remaining testis can increase its spermatogenic capacities subsequent to hemicastration.

## 5. Conclusions

This study describes, for the first time, the detailed recovery of testicular endocrine and germinative function after five months of treatment with a 4.7 mg deslorelin slow-release implant and subsequent removal. Our study confirmed the quick and full restoration of testosterone production and the reversibility of treatment-induced infertility with, however, a larger individual variation possibly related to differences in the full restoration of hypothalamus–pituitary–gonadal axis and/or Sertoli cell function. Even though first spermatozoa were present earlier (weeks 7–10), the breeding of previously treated male dogs should first be planned between weeks 12 and 19, preferably after a semen collection to confirm normospermia. Finally, as the data were obtained from Beagle dogs only, further studies investigating large populations of various breeds, body weights, and ages of dogs are required to gain deeper overall insights. Until then, individual observation of each clinical case is recommended.

## Figures and Tables

**Figure 1 animals-12-02545-f001:**
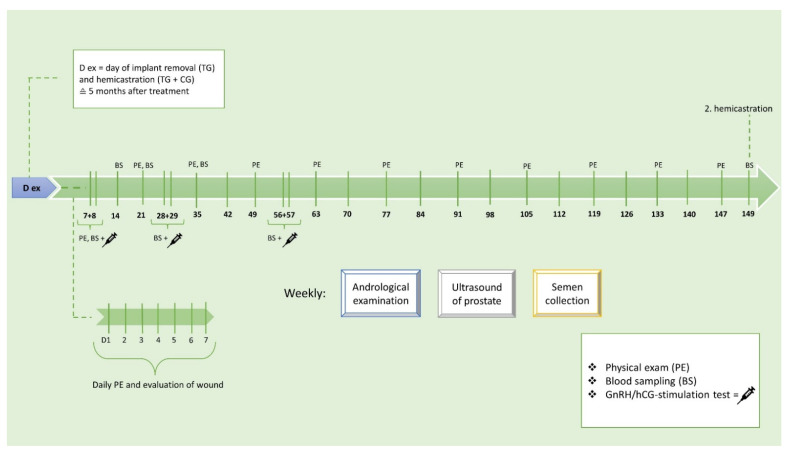
Detailed experimental design. The study includes two groups: TG animals (*n* = 7) were treated with a 4.7 mg deslorelin slow-release implant (SRI) in the paraumbilical area, whereas CG animals (*n* = 3) were subcutaneously treated with 0.5 mL saline in the umbilical area. In one TG animal (Lui), the induced effects were already fully reversed, which is why he was excluded from further analysis here.

**Figure 2 animals-12-02545-f002:**
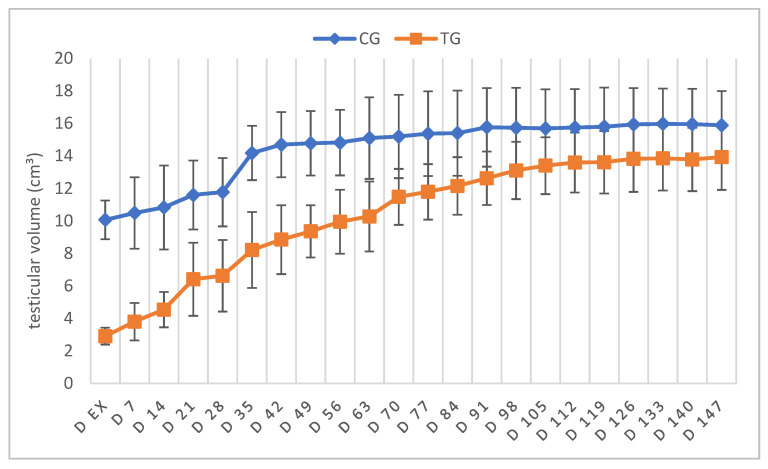
Absolute testicular volume measured by a caliper given as mean ± SD over time; animals treated with a 4.7 mg deslorelin implant (TG, *n* = 6); saline-treated control animals (CG, *n* = 3).

**Figure 3 animals-12-02545-f003:**
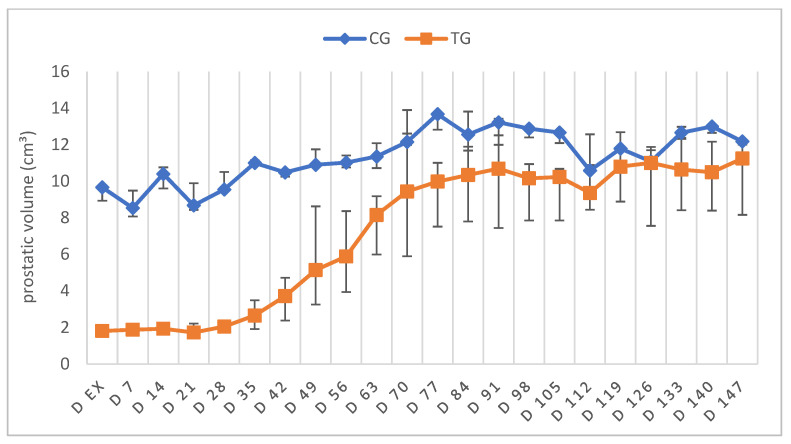
Absolute prostatic volume given as median (Q1, Q3) over time of animals treated with a 4.7 mg deslorelin implant (TG, *n* = 6) and saline-treated control animals (CG, *n* = 3).

**Figure 4 animals-12-02545-f004:**
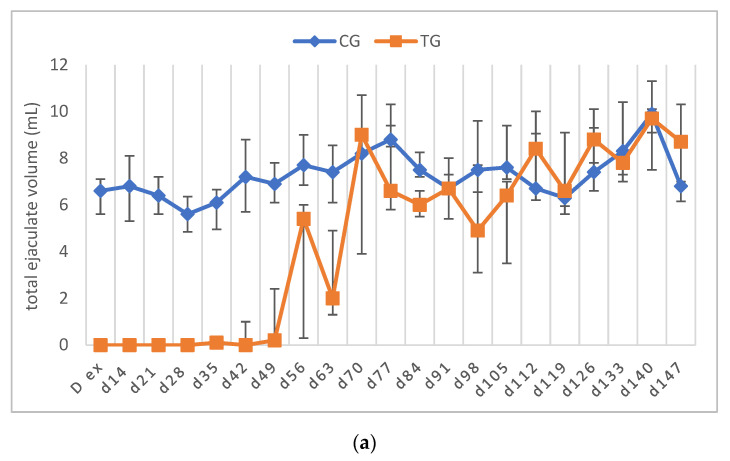
(**a**) Total ejaculate volume (mL) given over time as median (Q1, Q3) of animals treated with a 4.7 mg deslorelin implant (TG, *n* = 5, excluding Murphy) and saline- treated control animals (CG, *n* = 3). (**b**) Total ejaculate volume (mL) given as individual courses of animals treated with a 4.7 mg deslorelin implant (TG, *n* = 6) over time.

**Figure 5 animals-12-02545-f005:**
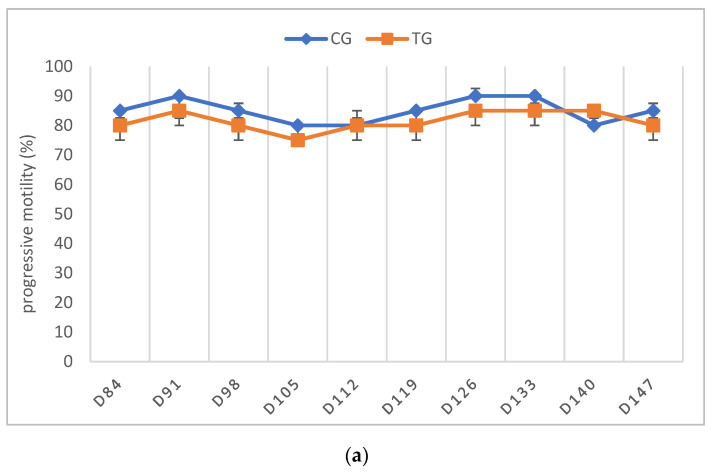
(**a**) Percentage of progressive motility given as median (Q1, Q3) over time of animals treated with a 4.7 mg deslorelin implant (TG, *n* = 5, excluding Murphy) and saline-treated control animals (CG, *n* = 3). (**b**) Percentage of progressive motility of spermatozoa given as individual courses of animals treated with a 4.7 mg deslorelin implant (TG, *n* = 6) over time.

**Figure 6 animals-12-02545-f006:**
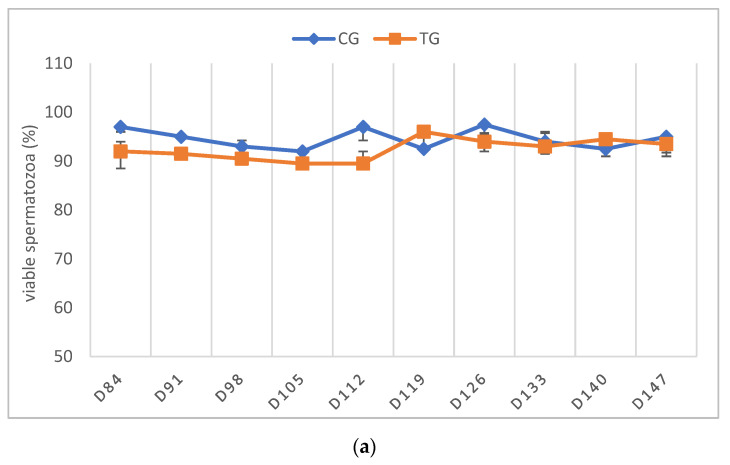
(**a**) Percentage of viable sperm given as median (Q1, Q3) over time of animals treated with a 4.7 mg deslorelin implant (TG, *n* = 5, excluding Murphy) and saline-treated control animals (CG, *n* = 3). (**b**) Percentage of viable sperm given as individual courses of animals treated with a 4.7 mg deslorelin implant (TG, *n* = 6) over time.

**Figure 7 animals-12-02545-f007:**
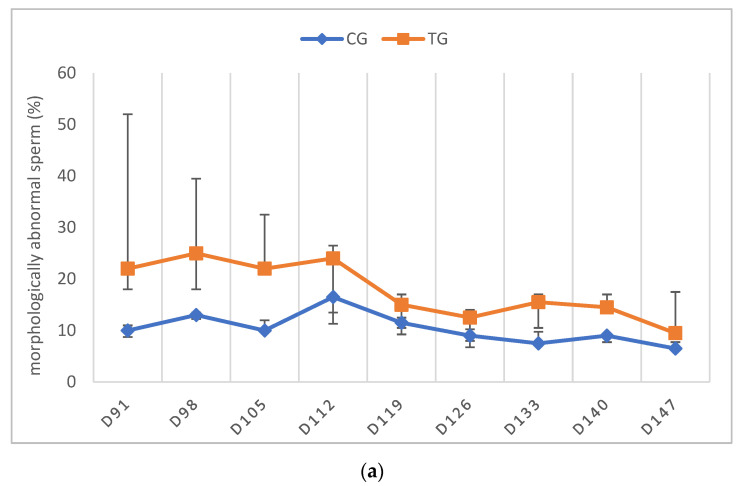
(**a**) Percentage of morphologically abnormal sperm given as median (Q1, Q3) over time of animals treated with a 4.7 mg deslorelin implant (TG, *n* = 5, excluding Murphy) and saline-treated control animals (CG, *n* = 3). (**b**) Percentage of morphologically abnormal sperm given as individual courses of animals treated with a 4.7 mg deslorelin implant (TG, *n* = 6) over time.

**Figure 8 animals-12-02545-f008:**
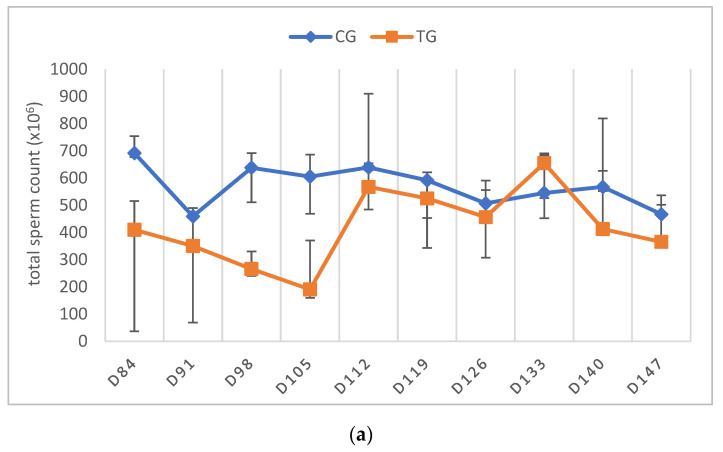
(**a**) Total sperm count (×10^6^) over time given as median (Q1, Q3) of animals treated with a 4.7 mg deslorelin implant (TG, *n* = 5, excluding Murphy) and saline-treated control animals (CG, *n* = 3). (**b**) Total sperm count (×10^6^) given as individual courses of animals treated with a 4.7 mg deslorelin implant (TG, *n* = 6).

**Figure 9 animals-12-02545-f009:**
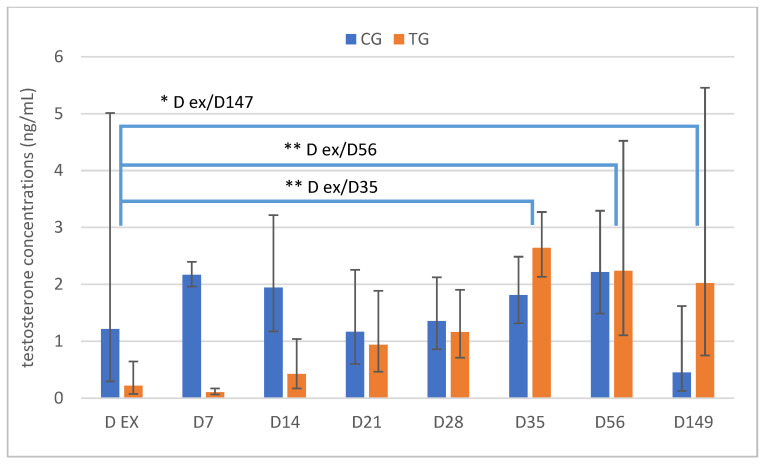
Plasma testosterone concentrations over time in ng/mL given as geometric mean ($\bar{x_{g}}$) and scatter range whose limits are defined by $\bar{x_{g}}$ x dispersion factor ± 1; * *p* < 0.05, ** *p* < 0.01; animals treated with a 4.7 mg deslorelin implant (TG, *n* = 6), and saline-treated, control animals (CG, *n* = 3).

**Figure 10 animals-12-02545-f010:**
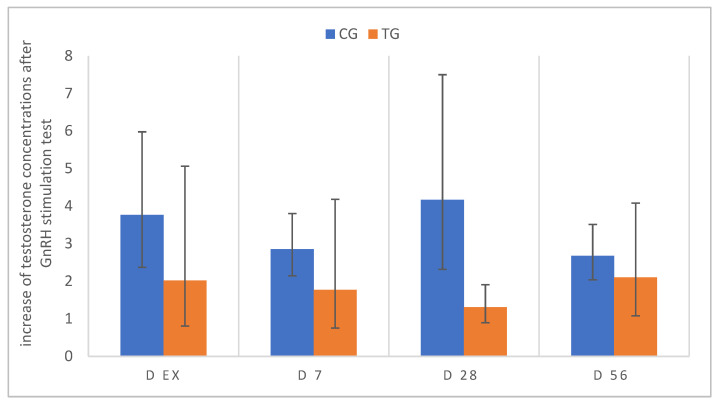
Relative increase in plasma testosterone concentrations 60 min after GnRH stimulation; pre-stimulation testosterone values are considered as 1, relative increase from pre- to post-stimulation is presented for D ex, D7, D28, D56; animals treated with 4.7 mg deslorelin implant (TG, *n* = 6); saline-treated control animals (CG, *n* = 3). Results are given as geometric mean $\bar{x_{g}}$ (DF).

**Figure 11 animals-12-02545-f011:**
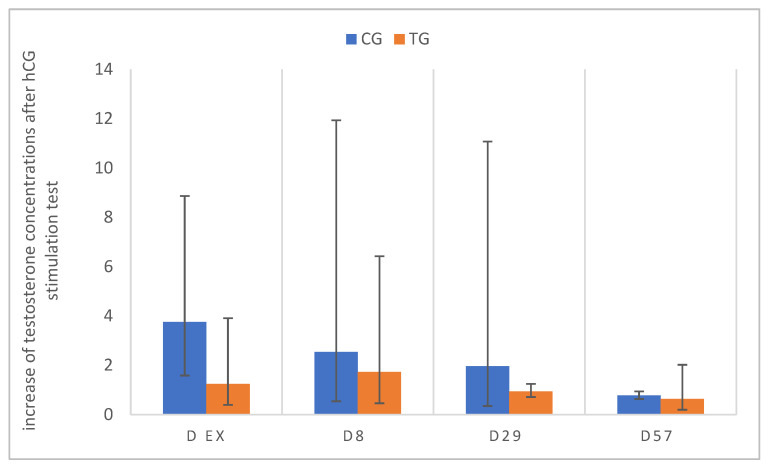
Relative increase in plasma testosterone concentrations 60 min after hCG stimulation; pre-stimulation testosterone values are considered as 1, relative increase from pre- to post-stimulation is presented for study day D ex, D8, D29, D57 animals treated with a 4.7 mg deslorelin implant (TG, *n* = 6) and saline-treated control animals (CG, *n* = 3). Results are given as geometric mean $\bar{x_{g}}$ (DF).

**Figure 12 animals-12-02545-f012:**
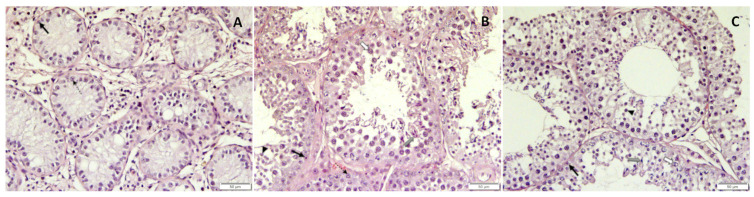
Histomorphology of testicular tissue after treatment with a deslorelin implant (**A**,**B**) or saline I. Whereas (**A**) represents an arrest of spermatogenesis on the level of mainly spermatogonia five months after treatment with the deslorelin implant (D ex, right testis), (**B**) clearly shows recovery with normal spermatogenesis being present at the end of the observational period (left testis). In (**C**), normal spermatogenesis of a saline-treated control at the end of the observational period is shown. 
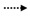
 spermatogonia; 
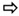
 spermatocytes; 
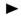
 round spermatids; 
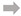
 elongating spermatids; 
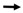
 Sertoli cells.

**Table 1 animals-12-02545-t001:** Comparative results of testicular volume (TV), prostate volume (PV), and testosterone (T) I. prior to treatment (D0), II. on D ex (before implant removal and hemicastration) and III. at the end of the observation period (recovery) for animals treated with a 4.7 mg deslorelin implant (TG, *n* = 6); results are given as mean ± SD (TV), median (Q1, Q3) (PV), and $\bar{x_{g}}$ (DF) (T).

	I	II	III
**Parameter**	**D0**	**D ex**	**Recovery**
**TV (cm^3^)**	8.6 ± 2.5 ^a^	2.9 ± 0.5 ^b^	13.9 ± 2.0 ^c^
**PV (cm^3^)**	9.5 (8.9, 11.6) ^a^	1.8 (1.7, 1.9) ^b^	11.3 (8.2, 12.0) ^a^
**T (ng/mL)**	3.8 (2.1) ^a^	0.2 (3.0) ^b,c^	2.0 (2.7) ^a,c^

Different superscripts indicate significant differences between time points.

**Table 2 animals-12-02545-t002:** Comparative results of pre-treatment semen analysis (I), of the first evaluable ejaculates after implant removal (II) and first normospermic ejaculates (III); results of TG (*n* = 5) are given as median (Q1, Q3).

Parameter	I	II	III
VOL (mL)	6.0 (2.3, 13.6)	6.6 (5.4, 10.3)	10.0 (6.0, 11.0)
% PM	65.0 (65.0, 75.0)	75.0 (75.0, 80.0)	80.0 (80.0, 85.0)
% VS	92.0 (91.5, 93.0)	93.0 (92.5, 95.0)	94.0 (93.0, 95.0)
% MAS	14.5 (14.0, 15.0) ^a^	79.0 (72.0, 86.0) ^b,c^	17.0 (15.5, 26.5) ^a,c^
TSC (×10^6^)	1091.6 (290.9, 1150.0) ^a,b^	56.3 (36.3, 62.5) ^a,b^	684.06 (567.19, 910.0) ^a^
days	Before	D56–91	D84–133

Different superscripts indicate significant differences between time points.

## Data Availability

The data presented in this study are available on request from the corresponding author.

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
