# Peer review of "What Happens in Male Dogs after Treatment with a 4.7 mg Deslorelin Implant? II. Recovery of Testicular Function after Implant Removal"

_animals, 2022, doi:10.3390/ani12192545_

Round 1
Reviewer 1 Report
This study focussed on the effects of using a GnRH agonist (Deslorelin), specifically the recovery of the testis following implant removal. The design of the study with 6 dogs in the treated group and 3 controls required a range of measurements to be taken before implant application, at downregulation before implant removal after 5 months treatment, and at recovery at D149. The study enabled histology through hemicastration at 5 months (implant removal) along with testicular and prostate measurements, semen analysis and hormone analysis. The left testis was then collected at Day 149. Of note in this study is that recovery was investigated at several levels: semen parameters, endocrine including (importantly) function tests through stimulation with buserelin and hCG.
The study is well conducted. It would have been stronger with a larger control control group however I feel that the within animal comparisons are performed well. The take home message arising from this study is that recovery does occur once the agonist is removed and that there is variation across animals. Despite the low number of animals, the authors even provide a recommendation for breeding previously treated dogs. My review points are therefore only of a minor nature.
1. I am not clear on is the histological observations made on the right testis. I would suggest the authors comment on this and provide images for comparison to the left testis collected at the end of the study.
2. Given that the one of the dogs (Murphy) was singled out on the basis of altered semen quality before treatment. I hadn't really appreciated this while reviewing the semen parameter graphs. This needs to be clarified in the appropriate legends.
2. Line 277: With regard to testosterone measurements, this increased in all TG dogs and was not different from the CG from D21 (not D14 as stated).
Author Response
Dear reviewer 1, thank you for your comments! Please find a letter attached
Sincerely
Sandra Goericke-Pesch

Reviewer 2 Report
Dear Authors,
The topic of this research is indeed interesting and information on the effect of deslorelin is needed and very actual in clinical practice. However, some doubts raised while reading your manuscript.
- This study is designed as a clinical trial (or laboratory trial); however, the sample size calculation is missing: are you sure that 9 dogs (because 1 was excluded) are sufficient to estimate any difference between the two groups?
- Were the dogs collected in the presence of a bitch in heath? With swabs? This can affect the libido and the volume of the ejaculate. Please detail this part.
- I would discuss that this is valid for a population on adult Beagle dogs, but this does not reflect a clinical situation. Would you state that regular spermatogenic function is expected to resume within 5 months from the removal regardless of the breed, size, age of the animal? Please add a comment on this in your discussion.
- Be careful to "Oxford commas" (e.g., line 101, line 124, 158, 161, 287, 342...).
Line 278: 'On the contrary, testosterone was not affected...'
Author Response
Dear reviewer 2, thank you for the review. Please find attached our response to your review.
Sincerely
Sandra Goericke-Pesch

Reviewer 3 Report
Although the cases included in the work are still few and making a correct statistic with 10 animals is not possible, the results presented correspond to the experience of those who work with deslorelin. The experimental data are well presented and correctly evaluated. the work can then be accepted in its present form.
However, I recommend continuing to collect cases to confirm the data obtained
Author Response
Thank you very much for your positive review. We are grateful for this.
All the best
Sandra
Round 2
Reviewer 2 Report
Dear Authors,
Thank you for answering the comments and changing the manuscript according to some of the suggestions.
If part I is considered together, the design and methods are clear. I think that your manuscript is suitable for publication. I encourage you to keep investigating this very interesting research topic.
Author Response
Thank you very much for your positive feedback. We appreciate this a lot!
All the best
Sandra Goericke-Pesch